# Accounting and Econometrics: From Paweł Ciompa to Contemporary Research

Marek Gruszczyński 

Institute of Econometrics, SGH Warsaw School of Economics, 02-554 Warszawa, Poland;
marek.gruszczynski@sgh.waw.pl

**Abstract:** This paper examines the little-known connection between econometrics and accounting invoked by Paweł Ciompa, who first introduced the term *econometrics* in 1910. Since then, research in accounting and in statistical (econometric) analysis has developed in parallel. It is argued that contemporary accounting research is methodologically closer to econometrics than ever before. This paper concentrates on the accounting origins of econometrics and on the econometric methodologies currently in use in accounting research, beginning with Paweł Ciompa's introduction of the term *econometrics* in accounting. The major contribution of this paper is a review of the occurrence of econometric methods in five leading journals in accounting research. The author identified 246 papers, and these were examined regarding the use of econometric methods. Two-thirds of the papers used methodologies that belong to econometrics—specifically, to financial microeconometrics. The most common methods were panel data models, qualitative variables models, and causality models.

**Keywords:** econometrics; accounting; financial microeconometrics; applied accounting

## 1. Introduction

This paper focuses on connections between econometrics and accounting. The first historical mention of the word *econometrics* was by Paweł Ciompa, a Polish economist, who was also a banker, teacher, and researcher in bookkeeping. In 1910, Ciompa had works published in Polish and in German that in English could have been entitled *Outline of Econometrics and Bookkeeping Theory*[1]. The Polish work was entitled *Zarys ekonometryi i teorya buchalteryi* (Ciompa 1910a) and the German book was *Grundrisse einer Oeconometrie und die auf Nationalökonomie aufgebaute natürliche Theorie der Buchhaltung* ( . . . ) Ciompa (1910b).

As used by Ciompa in 1910, the term *econometrics* has no relation to the word as used in modern economics. Ciompa's publications use it to present mathematically the rules of bookkeeping. However, he is recognized by many scholars as being the first to use the term in economics. Section 2, below, presents details of Ciompa's econometric ideas in accounting as well as the recognition of Ciompa by other scholars, including econometricians.

Today, the econometric methodology is a critical element in accounting research. To demonstrate this, Section 3 identifies the methodologies that appeared in papers in five major accounting journals in the period 2017–2021. Section 4 provides a conclusion. The contribution of this paper to the accounting literature lies in its underscoring of the connection between accounting and econometrics in contemporary research, as well as identifying the roots of this connection in the works of Polish scholar Ciompa, who published his thoughts more than 110 years ago.

## 2. The Econometrics of Ciompa

Paweł Ciompa (1867–1913) was a Polish banker, teacher, social worker, and the inventor of a new vocabulary of bookkeeping rules, for which he remains famous. While a

comprehensive exposition of Ciompa's theory is presented by Sojak (2022), the focus here is on the first-time use of the word *econometrics*.

Ciompa's idea of connecting accounting to econometrics was shown by Israel (2016). In his own translation, the author quotes the following from Ciompa (1910b): "Just like mechanical, acoustical, dynamic, and other such phenomena in physics, and mass phenomena in geometry, also economic phenomena should be represented and displayed following a doctrine, which I envision as a sort of economographics. This economographics would constitute a descriptive economics; it would have to be based on economics, mathematics, and geometry. The foremost task of such a doctrine would be the geometrical representation of value. This part of economographics I call econometrics. The practical application of econometrics to the mathematical representation of values and their changes would be accounting. Put differently, econometrics would then just be the theory of accounting".

Another translation of Ciompa (1910a) by Sojak (2022) states: "In theory, economics seeks to explain all phenomena of value, while in practice, mathematics, and bookkeeping account for the values of goods. Bookkeeping thus stands in a close relationship to economics. The theory of bookkeeping must be based on economics, and the rules of bookkeeping must be justified by economics. Just as physics represents mechanical, acoustic, dynamic, etc. phenomena, so too should economic phenomena be represented by the science we call econometrics. Econometrics is based on economics, mathematics, and geometry, and is part of economics, just as trigonometry is part of geometry. Bookkeeping is then only an application of econometrics, just as mathematics applies the laws of algebra".

Thus, econometrics according to Ciompa has its origin in geometry. Econometrics, through mathematics, forms the foundation of accounting. In a broader sense Ciompa is still right today: economics plus measurement = econometrics. The term for econometrics in Polish at that time was *ekonometrya* (now: *ekonometria*) while geometry in Polish was *geometrya* (now: *geometria*). Ciompa applies the Polish geometry ending *trya* to the new word, though there is no parallel in English.

The basic econometric equation by Ciompa (1910a) is the scheme:

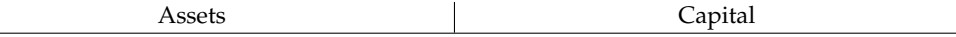

| Assets | Capital |
|---|---|

This graphical representation can be explained as follows: assets are something real—i.e., positive [+]—while capital is only its creative force ("econometric activity of assets")—i.e., something negative [−] (Sojak 2022). It should be noted that Ciompa's "capital" today also includes liabilities. Ciompa introduces "econometric equations and econometric ratios" through the graphical representations of rectangles. He labels such rectangle a "quadrigon", in Polish *kwadrant*. "Econometrically, the value of a good is the product of the quantity of the good and its unit price. Geometrically, this product is the area of a rectangle, one side of which has a length equal to the quantity of the good and the other to the unit price of the good (Ciompa 1910b, p. 10). Ciompa calls this rectangle the field (plane) of the value of the commodity" (after Sojak 2022). More details are provided in the paper by Sojak (2022).

The new wording and the geometric framework of Ciompa's proposal did not constitute a new theory. Lulek (1922) criticized Ciompa by stating that he "does not create new economic concepts and does not seek ways to solve economic problems" (Israel 2016, p. 3) (after Sojak 2022). In his own research on Ciompa's work, Israel (2016) declares that "his conceptions of econometrics and economographics are entirely descriptive. They are attempts to build upon economic theory, without transforming it. Thus, it stands in sharp contrast to the modern conception of econometrics".

In Poland, the legacy of Ciompa is evoked from time to time, both in accounting, as in Biadacz (2015) and Knop (2004), and in econometrics (statistics)—e.g., in Rutkowski (2009).

The current understanding of econometrics was defined 16 years after Ciompa by the first winner of the Nobel Prize in economics, Ragnar Frisch. In a Norwegian periodical, Frisch (1926) defined the new discipline of *econometrie*: "Intermediate between mathematics, statistics, and political economy, we find a new discipline, which, for lack of a better name,

may be called econometrics. It is the aim of econometrics to subject abstract laws of theoretical political economy or 'pure' economics to experimental and numerical verification, and thus to turn pure economics, as far as possible, into a science in the strict sense of the word" (Translation by Israel 2016).

The connection between Ciompa and Frisch has been acknowledged by Pesaran (1990), who admits: "The term 'econometrics' appears to have been first used by Pawel Ciompa as early as 1910; although it is Ragnar Frisch, one of the founders of the Econometric Society, who should be given the credit for coining the term, and for establishing it as a subject in the sense in which it is known today". In a note in *Econometrica*, Frisch stated that he was not aware of Ciompa's work (Frisch 1936; Israel 2016).

The story of Ciompa and his novel term "econometrics" continues today. Accounting research over many years has been intertwined with econometrics. Methodology pertaining to econometrics—specifically to microeconometrics—is increasingly present in accounting research. The following section presents a review of papers published recently in renowned accounting journals worldwide.

## 3. Contemporary Accounting Research and Econometrics

### 3.1. Selection of Accounting Journals for Review

More than 110 years after publication of Ciompa's book, this research attempted to discover how accounting and econometrics are jointly present today in journal publications. Obviously, the econometrics methodology seen in current accounting research is modern econometrics, a term coined by Ragnar Frisch. Today, Ciompa's econometrics has only historical value. Furthermore, the topics of contemporary accounting research are far removed from those discussed at the beginning of the twentieth century.

Initially, five leading journals in accounting research were selected from the list of respected journals presented in Gruszczyński (2022). The sources for that list were the Web of Science (Social Science Citation Index and Emerging Sources Citation Index) and the Scopus list of the major journals in the field of "Business, Management and Accounting" by country, as published by Scimago. Considered were the following 21 journals:

- *European Accounting Review, Accounting in Europe*; journals of the European Accounting Association (EAA)
- *The Accounting Review, Journal of Management Accounting Research*; journals of the American Accounting Association (AAA)
- *Contemporary Accounting Research*; journal of the Canadian Academic Accounting Association
- *Journal of Accounting Research*
- *Journal of Accounting and Economics*
- *Journal of Business Finance and Accounting*
- *Journal of International Financial Management and Accounting*
- *Review of Accounting Studies*
- *Journal of Accounting Literature*
- *Journal of Accounting Auditing and Finance*
- *International Journal of Accounting Information Systems*
- *Critical Perspectives on Accounting*
- *British Accounting Review*
- *Advances in Accounting*
- *Accounting Horizons*
- *Abacus—A Journal of Accounting Finance and Business Studies*
- *Accounting and Finance*
- *Comptabilite Controle Audit*; journal of the Francophone Association of Accounting
- *Revista de Contabilidad-Spanish Accounting Review*; journal of the Spanish Association of Accounting Academics.

The five journals finally selected for examination in this review are presented in Table 1. They were chosen somewhat arbitrarily, but they are all renowned, frequently cited journals

with high impact factors, and are representative of advanced accounting research published in the USA, Canada, and Europe. Articles published in these journals during the five-year period 2017–2021 were examined.

**Table 1.** Accounting journals examined in the paper for the period 2017–2021.

| Journal Title | Editor/Publisher | IF2021 |
|---|---|---|
| *European Accounting Review* | European Accounting Association/Taylor & Francis | 4.761 |
| *Contemporary Accounting Research* | Canadian Academic Accounting Association/Wiley | 4.446 |
| *Journal of Accounting Research* | Chicago Booth School of Business/Wiley | 7.293 |
| *Journal of Accounting and Economics* | Elsevier | 2.845 |
| *The British Accounting Review* | British Accounting and Finance Association/Elsevier | 4.041 |

In each of the 5 years, a single issue was selected from each journal for a total of 25 issues. All papers published in those 25 issues were reviewed, 246 papers in total. In accordance with the focus of this research, the papers were examined for the use of econometric methodology.

*3.2. Econometric Methods in the Selected Accounting Journals*

The 25 selected issues were examined for papers applying econometric and other quantitative methods. The papers' topics were also recorded and categorized. Table 2 presents for each journal a list of the issues examined, the number of papers published, the number of papers using any quantitative method, and the number of papers using at least one econometric method.

**Table 2.** Accounting journals, the list of issues, the number of papers using quantitative/econometric methodology.

| Issues | Number of Papers Published | Number of Papers Using Any Quantitative Method | Of Which: Papers Using Econometric Method(s) |
|---|---|---|---|
| *British Accounting Review* <br> 2017 vol. 49 no. 1, 2018 vol. 50 no. 3, <br> 2019 vol. 51 no. 5, 2020 vol. 52 no. 4, <br> 2021 vol. 53 no. 2 | 34 | 12 | 11 |
| *Journal of Accounting Research* <br> 2017 vol. 55 no. 1, 2018 vol. 56 no. 3. <br> 2019 vol. 57 no. 5, 2020 vol. 58 no. 4, <br> 2021 vol. 59 no. 2 | 32 | 32 | 31 |
| *Journal of Accounting and Economics* <br> 2017 vol. 63 no. 1, 2018 vol. 66 no. 3, <br> 2019 vol. 68 no. 2–3, 2020 vol. 70 no. 2–3. <br> 2021 vol. 71 no. 2–3 | 50 | 48 | 46 |
| *European Accounting Review* <br> 2017 vol. 26 no. 1, 2018 vol. 27 no. 3, <br> 2019 vol. 28 no. 5, 2020 vol. 29 no. 4, <br> 2021 vol. 30 no. 2 | 36 | 30 | 24 |
| *Contemporary Accounting Research* <br> 2017 vol. 34 no. 1, 2018 vol. 35 no. 3, <br> 2019 vol. 36 no. 1, 2020 vol. 37 no. 4 <br> 2021 vol. 38 no. 2 | 94 | 85 | 60 |
| *Total* | 246 | 207 | 165 |

One major finding of the research is that two-thirds of all the papers examined in the selected issues published by the five accounting journals during each of the past five years used econometric methods. This is again evidenced in Table 3, where the split between

non-econometric quantitative methods are divided between methods of mathematical economics and other methods. Research based on mathematical economics is presented in 9% of papers, while 8% of papers use other quantitative methods.

**Table 3.** Summary of the survey's main findings.

|  | Number of Papers | Percent of the Total |
|---|---|---|
| Total number of papers published | 246 | 100% |
| Number of papers that use any quantitative method | 207 | 84% |
| of which: |  |  |
| papers using econometric method(s) | 165 | 67% |
| mathematical economics papers | 23 | 9% |
| papers using other quantitative methods | 19 | 8% |

The investigation shows that the accounting research published in these respected journals employs several econometric methodologies. Table 4 shows that the most common methods applied are:

– Panel data models—constituting nearly 80% of all econometric papers;
– Models of qualitative variables, binomial and multinomial—30% of econometric papers;
– Causality models—20% of econometric papers.

**Table 4.** Numbers of papers using specific econometric methods.

| Papers Using Econometric Method(s) | 165 |
|---|---|
| More than one method applied—44% of econometric papers | 73 |
| Regression—cross section/time series (no panel approach): returns (Fama-MacBeth), survey data, etc. | 24 |
| Regression/time series (event analysis, finance) | 6 |
| Panel data models—78% of econometric papers | 129 |
| Models of qualitative variables: binomial (logit/probit/LPM) also panel Data approach | 40 |
| Models of qualitative variables: multinomial | 9 |
| Model of limited-dependent variables (tobit) | 1 |
| Models of causality: treatment effects (PSM, RDD, diff-in-diff) | 29 |
| Count data model | 1 |
| Sample selection (Heckman) | 7 |

The total exceeds 100% because many papers (44%) applied more than one econometric method.

Nearly all methodological approaches in the examined papers belong to financial microeconometrics. This area of econometrics represents the microeconometrics and metrics applied to corporate finance and accounting (as presented in the book by Gruszczyński 2020). The sets of microdata used for such research are typically financial data from companies over space and time.

The most common methodology is panel data econometrics. The typical structure of the research part of the paper is as follows:

– Data are carefully collected, usually from popular databases, as firm-years observations;
– Variables are designed along with the research theme and the formulated hypotheses, with reference to previous research in the area;
– Descriptive statistics are presented (parameters of variables' distributions), along with Pearson correlation coefficients;
– Explanatory variables are then selected for several variants of the designed model;
– Usually, linear panel models with fixed effect (firm/year) are applied;

– Finally, the estimation results are presented and interpreted.

Many papers also examine the question of endogeneity, using the instrumental variables approach. Often the panel model is accompanied by other approaches—e.g., bivariate model of qualitative variable, analysis of treatment effects, etc. Models of qualitative variables usually appear in the papers along with other models. These include logit and probit models, both binomial and multinomial (ordered and unordered). Causality models use propensity score matching, diff-in-diff, regression discontinuity design, natural or quasi-natural experiment, etc.

### 3.3. Accounting Topics Studied in the Examined Papers

Contemporary accounting research covers a wide range of topics, including those typically pertinent to corporate finance or corporate governance, as confirmed by the review. The list of topics from the 25 papers selected from the surveyed journals is presented here.

– Earnings announcement disclosures;
– Tax preparation expenses;
– Director external social networks and crash risk;
– Analysts' earnings expectation management;
– Archival evidence on the audit process;
– Labor unions and income smoothing;
– Creative culture and real earnings management;
– Tone management in the management discussion and analysis report;
– Audit regulation and the cost of equity capital;
– Do corporate site visits impact stock prices?
– Audit firm tenure, bank complexity, and financial reporting quality;
– Earnings management and CEO marital status;
– Financial statements—tool for monitoring borrowers;
– Auditor social and human capital—the effect on auditor compensation;
– Gambling attitudes in financial misreporting;
– Litigation risk and corporate voluntary disclosure;
– Voluntary disclosure and stock liquidity;
– Retaliation costs and employee whistleblowing;
– Mutual fund investors and auditor quality;
– Earnings announcement premium;
– Opacity and the cost of debt for family firms;
– Size management to minimize the cost of disclosure;
– Effects of financial reporting and disclosure on corporate investment;
– Cross-border migration and the accounting profession;
– Shareholder litigation and corporate disclosure;
– Product information on Twitter and firm sales;
– Financial reporting quality and tick size (natural exp);
– Female directors and earnings management;
– The role of auditing in the fight against corruption;
– Country-level corruption and accounting choice.

The topics and research questions listed above represent accounting, corporate finance, corporate governance, and economics. This confirms the broad scope of the articles in the accounting journals chosen for this survey.

### 3.4. Econometric Research in Accounting—Four Examples from the Survey

To underscore the wide range of topics and approaches among the surveyed journals, summaries of four papers using various econometric methodologies are presented here.

### 3.4.1. Multiple Regression

Geoffroy and Lee (2021) examined how academic research is consumed by the Securities and Exchange Commission (SEC), especially before and after the court decision in the case of *Business Roundtable* vs. SEC in 2011. The authors studied cost–benefit analyses by the SEC (performed before the introduction of new regulations) between 2007 and 2017. The primary dataset consisted of 176 proposed and 181 final SEC regulation releases that contained 738 (proposed) and 389 (final) citations of academic research. The authors also collected commented letters on the proposed SEC releases and used Amazon M-Turk to classify them into positive, negative, and neutral. Two surveys were conducted: the first limited to the authors of papers cited in the proposed and final SEC regulations, and the second distributed to the larger academic community. In addition, the number of citations, the tone of citations, etc. were identified and collected.

Several OLS regression models are presented in the paper. For example, the first model is as follows:

$$Presence(Number)\ of\ Citations = \beta_0 + \beta_1 Post + \beta_2 Final + \beta_3 Post \times Final + \beta_4 Length\ of\ Document + \beta_5 Mandatory + \varepsilon$$

with two possible explained variables: the presence of citations in a document (the dummy variable equals 1 if at least one citation occurs and equals 0 otherwise) and the number of citations in a document. Explanatory variables are: Post, the dummy variable takes the value of 1 if the document was written after the *Business Roundtable* case (March 2012) and 0 otherwise; Final, the dummy variable takes the value of 1 if the document is a final rule and 0 if it is a proposed rule; *Length_of_Document*, the natural logarithm of the total word count in the document; *Mandatory*, the dummy variable which takes the value of 1 if the rule was written as part of the Dodd–Frank Act or the JOBS Act, and 0 otherwise.

Two other regressions estimated in this paper are the regression of *Citation_Tone* (with the same explanatory variables as in the first regression above) and the regression of *Percentage_of_Negative_Comment_Letters*, with explanatory variables also including the number, occurrence, and tone of citations.

The major finding of this paper is that, after the court decision in 2011, the SEC cited more papers in its proposed rules, particularly papers that show the costs of regulation. Additionally, the number of negative comments on proposed SEC regulations was lower after 2011.

### 3.4.2. Panel Data Model

Bonacchi et al. (2018) investigated the question whether Italian nonlisted subsidiaries engage in earnings management so that their listed parent companies can meet benchmarks. The sample of companies covers the period from 2003 to 2014. Two ways of managing earnings in subsidiaries are considered: accrual and real earnings management.

The variable representing accrual-based earnings management in subsidiaries is calculated with the use of coefficients from the model of normal accruals that is estimated for a sample of stand-alone companies (not subsidiaries). Similarly, real earnings management in subsidiaries is represented by the variable calculated with the use of coefficients from the model of normal cash flow from operations that is estimated for a sample of stand-alone companies.

The authors also identified "suspect firm-years" when earnings are likely to have been managed. There are three benchmarks that a firm is likely to meet or beat: zero earnings, previous year's earnings, and analysts' forecasts. A parent company is considered suspect in years when (1) it reports a small profit, (2) it reports a minor change in profits, and (3) it reports positive analyst forecast errors.

The primary panel regression is as follows:

$$Y_{it} = \alpha_0 + \beta_1 SIZE\_PC_{it} + \beta_2 \Delta S\_SUB_{it} + \beta_3 EBXI\_SUB_{it} + \beta_4 Suspect\_PC_{it} + \beta_5 Suspect\_EBXI\_SUB_{it} + \sum_{i=1}^{n-1} \delta_i FirmFE_i + \sum_{t=1}^{T-1} \gamma_t YearFE_t + \varepsilon_{it}$$

where $Y_{it}$ represents discretionary accruals or abnormal cash flow from operations, *PC* denotes parent company, and *SUB* denotes subsidiary. *Suspect_PC$_{it}$* is a dummy variable taking the value of 1 when, alternatively, the parent company is considered suspect in the years type (1) or (2) or (3) above. Variable $SIZE\_PC_{it} = Asset\_PC_{it} / Asset\_SUB_{i,t-1}$. Variable $\Delta S\_SUB_{it}$ is change in sales and variable $EBXI\_SUB_{it}$ is return on assets, both variables being scaled by the subsidiary's lagged assets. $Suspect\_EBXI\_SUB_{it}$ is a dummy variable taking the value of 1 when the subsidiary's $EBXI\_SUB_{it}$ is within a small specified range. Variables representing firm and year fixed effect are also included.

The estimation results of the above model yielded evidence that suspect parent companies use their subsidiaries to manage their consolidated earnings to avoid losses and to beat analyst forecasts. There was no evidence that Italian parent companies use subsidiaries to beat previous year's earnings.

The authors also estimated other panel regression models of $Y_{it}$—e.g., with variables representing Big 4 auditors of suspect companies and with variables indicating if the percentage of parent directors also holding a position on the subsidiary's board exceeds 50%. The results showed that Big 4 auditors at the level of parent company mitigate accrual earnings management at the level of the subsidiary, and that parent companies also coordinate earnings management through parent directors holding positions on the subsidiary's board.

### 3.4.3. Qualitative Variable Model

Bernard et al. (2018) studied size management by European private firms for which disclosure requirements increase when hitting a size threshold. Data on private limited liability companies in 12 European countries constitute 503,666 firm-year observations from the period 2003–2011. There are size-variable observations within 2% of the thresholds. The authors estimated several binomial logit models, beginning with the following:

$$Below\_threshold_{it} = \gamma_0 + \gamma_1 Expanded\_disclosure_{it} + \gamma_2 External\_audit_{it} + \varepsilon_{it}$$

This model was estimated both with country-year fixed effects (panel binomial regression) and without those effects. The dependent variable *Below_threshold* is the proxy for size management that equals 1 if the observation is in the bin immediately below the threshold and 0 if it is immediately above, where bin size is 2% of the threshold. *Expanded_disclosure* equals 1 if expanded public disclosure requirements are imposed at the threshold to which the observation is adjacent and 0 otherwise. *External_audit* is equal to 1 if a mandatory audit requirement is imposed at the threshold to which the observation is adjacent, and 0 otherwise.

Successive logit models included additional regressors, like *Income_statement_disclosure* (0–1 variable), *Cash_flow_statement_disclosure* (0–1 variable) and others. The report of estimation findings concentrates on the significance of estimated coefficients, without exploring the details of prediction and classification results.

The major result of the research is that European private companies manage the size downward at size thresholds that impose expanded disclosure, particularly income statement disclosure. The authors show that at least 8% of firms that would otherwise be immediately above a size threshold manage size to avoid income statement disclosure. They also estimate that "the costs of public income statement disclosure are substantial enough to lead firms that manage size to sacrifice, on average, roughly 6.5% of their asset size, which corresponds to approximately 7–9% of income".

### 3.4.4. Causality Model

Sultana et al. (2019) used several models in their research, including the method of diff-in-diff (DiD). One of the research questions was whether audit fees change after the appointment of a new audit committee member (outside the firm) with different experience levels to the outgoing audit committee member. The authors used firm-year observations on Australian companies listed on the ASX for the period 2001–2012.

The DiD methodology compared audit fees before and after switches from an experienced audit committee member to a less experienced audit committee member (treatment group)—with a control group of switches at the same level of experience. This was achieved by testing whether audit committee members that were substituted with others having more multiple directorships and being older resulted in an increase in audit fees. The following models were estimated for the pooled sample (both control and treatment groups):

$$\Delta LnAfees_{it} = \beta_0 + \beta_1 Less\_to\_More\_AC\_Mul_{it} + \gamma_j \Delta Control\_variables_{it} + \varepsilon_{it}$$

$$\Delta LnAfees_{it} = \beta_0 + \beta_1 Younger\_to\_Older\_AC\_Age_{it} + \gamma_j \Delta Control\_variables_{it} + \varepsilon_{it}$$

where the explained variable is the change of $LnAfees_{it}$—the logarithm of total audit fees paid by firm $i$ to its auditor for audit services in time period $t$. Explanatory variables included the dummy variable $Less\_to\_More\_AC\_Mul_{it}$ that equals to 1 if a new audit committee member joining the firm had a greater number of multiple-directorships than the exiting audit committee member, and the dummy variable $Younger\_to\_Older\_AC\_Age_{it}$ that equals to 1 if the new audit committee member joining the firm was older than the exiting audit committee member.

Results indicated that firms, having appointed an audit committee with more multiple-directorships and older audit committee members, pay higher audit fees. This indicates a causal effect of greater audit committee member experience on audit fees.

## 4. Conclusions

Accounting research has come a long way between the publication of Paweł Ciompa's book over 100 years ago and the accounting papers published today in renowned journals. In this paper, the connection was examined between econometrics as first defined by Ciompa in the context of accounting in 1910, and modern econometrics used in contemporary accounting research. Ciompa's econometrics has no connection to the term econometrics proposed by Ragnar Frisch in 1926 to describe the methodology of statistics and mathematics in economics. Therefore, the oldest use of *econometrics* now has only historical and, for some, sentimental value. In contrast, modern econometrics is very much alive and present in contemporary accounting research.

Our research examined 246 papers published in five accounting journals of international reputation. The major outcome of this survey is that 84% of papers examined in these journals in the past five years used quantitative methods. Moreover, two-thirds of all papers (67%) examined used econometric methods. These methods mostly belong to financial microeconometrics (Gruszczyński 2020). A more detailed breakdown of results shows that the most common were panel data models, qualitative variables models, and causality models.

This review presents research published in a group of renowned international journals, representing the current mainstream of modern accounting research. Other journals may not follow the extent and depth of econometric methodology as seen in the selected journals here, instead focusing more typically on the practical side of accounting, including from legal or managerial perspectives.

**Funding:** This research received no external funding.

**Conflicts of Interest:** The author declares no conflict of interest.

## Note

1     Translation by Sojak (2022).

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
