# Peer review of "Accounting and Econometrics: From Paweł Ciompa to Contemporary Research"

_jrfm, doi:10.3390/jrfm15110510_

Round 1
Reviewer 1 Report
Thank you for the opportunity to read and review your paper entitled “Accounting and econometrics: from PaweÅ‚ Ciompa to contemporary research”.
In my opinion, the topic is on time and the paper can contribute to the current academic debate. I hope you will find the comments useful to further improve the paper.
I look forward to receiving your revised manuscript soon.
Title
The title, “Accounting and econometrics: from PaweÅ‚ Ciompa to contemporary research” expresses at its best the content of the paper.
Abstract
The abstract outlines in short the aim, the theory and results which increase reader’s interest in keep reading the contribution.
Originality: Does the paper contain new and significant information adequate to justify publication?
The paper addresses an important research question which is under the current academic debate. Hence, I believe that the paper can significantly contribute to the extant literature.
Introduction
In my opinion, the “Introduction” section is too short. The author(s) should enlarge it better describing how the current paper contributes to the extant literature.
Contemporary accounting research and econometrics
The “Contemporary accounting research and econometrics” section is generally well organised.
However, I ask to the author(s) to improve this section by providing more insights on recent articles on the same topic published in this journal, if available.
Methodology
The “Data and methodology” section is well executed.
However, I suggest the following improvements.
When describing the data and methodology, the author(s) directly reported the results in the tables. I think it is better to also report these results through graphs and figures, e.g., pie-charts or similar. It could improve the reader’s interest.
Practicality and/or Research implications
The author(s) should evaluate the opportunity to stress further the implication for practice and academic world, since they appear to be too weak.
Quality of Communication
The overall quality of the paper is adequate, since the author(s) used the proper technical language of the field. However, it is possible to notice a strict separation among the various section of the paper. Although the structure of the paper is clear, the logical steps that lead from a section to another section are not clear too. The author(s) should try to better link the various sections of the manuscript.
In addition, I recommend to author(s) to check carefully potential grammar errors before to submit the revised manuscript, as well as to make the writing more fluid, careful and clean. The use of a proofreading service is strongly recommended.
Author Response
Response to reviewer 1 is attached in the for of Word file.

Reviewer 2 Report
The paper is quite interesting but some improvements are needed. It has to be emphasized that this paper is a professional paper. There are no research questions nor research hypotheses. There is no real scientific contribution. Only review of a certain number of papers is conducted in the paper as a part of the paper. Please write in the third person in a passive form. "Basic econometric equation" or balance sheet equilibrium should be that assets' value is equal to capital plus liabilities (short- and long-term) values. It is unclear how you selected five journals. First you write about five journals, then you put a list of 21 journals and then you reduce it to five by unclear criteria. Why in Table 2 are not listed all issues of the observed journals? For example, you have Vol. 49, No. 1 and the following is Vol. 50, No. 3. What do you consider under "econometric methods"? That needs further explanation. Maybe you should replace the word "survey" with some other more appropriate word like research or investigation or something similar.Author Response
Response to reviewer 2 is attached in the Word file.
